# Fourteen-Day Gemcitabine-Docetaxel Chemotherapy Is Effective and Safer Compared to 21-Day Regimen in Patients with Advanced Soft Tissue and Bone Sarcoma

**DOI:** 10.3390/cancers13081983

**Published:** 2021-04-20

**Authors:** Minggui Pan, Maily K. Trieu, Manpreet Sidhu, Jeanette Yu, Tiffany Seto, Kristen Ganjoo

**Affiliations:** 1Department of Oncology and Hematology, Kaiser Permanente, Santa Clara, CA 95051, USA; 2Division of Research, Kaiser Permanente, Oakland, CA 94612, USA; 3Department of Drug Utilization, Kaiser Permanente, Oakland, CA 94612, USA; Maily.K.Trieu@kp.org; 4Department of Oncology and Hematology, Kaiser Permanente, Roseville, CA 95661, USA; manpreet.sidhu@kp.org; 5Department of Oncology and Hematology, Kaiser Permanente, Oakland, CA 94611, USA; jeanette.c.yu@kp.org; 6Oncology and Hematology Fellowship Program, Kaiser Permanente, San Francisco, CA 94115, USA; tiffany.seto@kp.org; 7Division of Oncology, Stanford University, Stanford, CA 94305, USA; kganjoo@stanford.edu

**Keywords:** sarcoma, gemcitabine, docetaxel, survival, genomic alterations

## Abstract

**Simple Summary:**

Gemcitabine-docetaxel chemotherapy is an important regimen for the treatment of soft tissue and bone sarcomas. We aimed to determine if gemcitabine-docetaxel when administered every 14-day would be as effective and less toxic compared to the 21-day schedule. Our study shows that indeed when administered in 14-day schedule gemcitabine-docetaxel chemotherapy results in similar chance of tumor shrinkage and survival yet fewer febrile neutropenia and discontinuation of chemotherapy due to intolerance, compared to 21-day schedule. Therefore, 14-day gemcitabine-docetaxel chemotherapy is safer and can be broadly adopted for the treatment of advanced soft tissue and bone sarcomas.

**Abstract:**

Gemcitabine-docetaxel (G-D) combination is an effective chemotherapy for patients with advanced soft tissue and bone sarcoma, first developed with G administered on days 1 and 8, and D on day 8 every 21 days and later modified to be administered every 14 days in 2012. The 14-day regimen has become increasingly adopted. However, its efficacy and toxicities have not been compared. We identified 161 patients with metastatic or locally advanced soft tissue and bone sarcoma treated with either a 14-day or 21-day regimen within Northern California Kaiser Permanente from 1 January 2017 to 30 July 2020 and compared the outcomes and toxicity profiles of patients treated with the either regimen. Seventy-nine (49%) and 82 (51%) patients received the 14-day and the 21-day regimen, respectively, with similar response rate (22.8% and 15.8%, *p* = 0.26), median progression-free survival (PFS, 4.0 and 3.2 months, *p* = 0.15), and median overall survival (OS, 12.6 and 14.7 months, *p* = 0.55). Subset analysis of the untreated patients (approximately 60% of the entire cohort) as well as the patients with leiomyosarcoma only (approximately 50% of the entire cohort) showed that OS was not significantly different between the two regimens. Febrile neutropenia requiring hospitalization occurred in 10 and one patients (*p* = 0.006) and intolerance leading to discontinuation of chemotherapy occurred in 12 and two patients (*p* = 0.006) treated with the 21-day and the 14-day regimens, respectively. CDKN2A deletion/mutation or CDK4 amplification was associated with worse median OS (*p* = 0.06), while a RB1 deletion/mutation was associated with better median PFS (*p* = 0.05), and these two genomic alterations were mutually exclusive. Our data demonstrate that, compared to the traditional 21-day G-D regimen, the 14-day G-D regimen is equally effective but safer. In addition, CDKN2A and RB1 pathways play significant role on the outcomes of the patients.

## 1. Introduction

Systemic therapy options for metastatic and locally advanced soft tissue (STS) and bone sarcoma are limited. Doxorubicin as single agent or in combination with ifosfomide and/or dacarbazine was considered standard, first-line chemotherapy [1,2], until Hensley and colleagues reported outcomes with fixed-dose gemcitabine (G) 900 mg/m^2^ administered at days 1 and 8, docetaxel (D) 100 mg/m^2^ at day 8, with granulocyte colony-stimulating factor (GCSF) support in 34 patients with advanced leiomyosarcoma (LMS) and showed a 53% overall response rate (RR) [3]. The same regimen in advanced uterine LMS patients showed an overall RR of 35.8% in a first-line and 27% in a second-line setting, in two phase II Gynecologic Oncology Group (GOG) trials [4,5]. A separate phase II trial from the United Kingdom showed a 25% RR in patients with metastatic uterine LMS [6]. This regimen was quickly adopted for treating other types of sarcoma as well. A retrospective analysis of 133 patients (50% non-LMS) treated with this regimen showed an 18.4% RR in all cohorts, and 24% for the non-LMS histologies [7]. The phase II Sarcoma Alliance for Research through Collaboration study (SARC002) showed that the G-D regimen was superior to single-agent G in RR and overall survival (OS) in patients with metastatic STS who had zero to three prior lines of chemotherapy, however, with higher rate of discontinuation due to toxicities [8]. This trial further established the G-D combination as an effective regimen for advanced STS and led to its broad adoption, despite the fact the TAXOGEM phase II French trial showed that single-agent G resulted in similar efficacy compared to G-D but with lower toxicity in patients with advanced leiomyosarcoma [9]. In the subsequent G-D trials, D dose was reduced from 100 mg/m^2^ to 75 mg/m^2^ [10,11,12,13]. It is worth noting that D as a single agent showed limited activity in STS except for angiosarcoma [14,15,16].

Verschraegen et al. modified the G-D dose and schedule and added bevacizumab (Axtell regimen) in a phase IB trial in patients with advanced/recurrent STS, using a 30-min infusion of G 1000 mg/m^2^, escalated to 1250 mg/m^2^ and 1500 mg/m^2^, and D 50 mg/m^2^, given every 14 days [17]. This regimen showed a RR of 31.4%. Bevacizumab is no longer combined due to lack of added efficacy in subsequent randomized trials [18]. The Axtell regimen without bevacizumab has been adopted by many oncologists due to its potentially lower toxicity and its convenience of administration. However, it is unclear if the efficacy of this regimen is equivalent to the every-21-days regimen and if its toxicity profiles are significantly different.

We performed a retrospective study on patients with metastatic or locally advanced STS and bone sarcoma treated with the 14-day versus the 21-day G-D regimen within Kaiser Permanente Northern California (KPNC) from 1 January 2017 to 30 July 2020. Our results showed that the 14-day G-D regimen was equally efficacious compared to the 21-day G-D regimen with similar OS, while being better tolerated. In addition, few studies have explored the association of specific genomic alterations with STS and bone sarcoma [19,20,21]. Bui et al. showed that genomic alteration of the CDKN2A pathway was associated with worse prognosis in patients with STS [19]. In this study we also explored the association of genomic alterations with outcomes and found that the patients with the genomic alteration of CDKN2A pathway were associated with worse OS. In addition, we found that RB1 deletion/mutation was associated with better PFS. We also found that genomic alteration of CDKN2A pathway and RB1 deletion/mutation were mutually exclusive.

## 2. Patients and Methods

### 2.1. Data Collection

We identified patients with metastatic or locally advanced STS and bone sarcoma who were treated with G-D regimen from 1 January 2017 to 30 July 2020 within KPNC, by mining the electronic database (Epic). KPNC is an integrated healthcare system that cares for approximately 4.5 million members in 21 medical centers. We chose this time frame because D dose was changed from 100 mg/m^2^ to 75 mg/m^2^ in 2017 in a majority of the patients treated with the 21-day regimen and that the 14-day regimen was becoming more widely adopted in our institution. The treatment regimen was determined by the patient’s treating oncologist. This study was approved by the KPNC Institutional Review Board.

### 2.2. Gemcitabine, Docetaxel, and GCSF Administration

For the 21-day regimen, G was administered at a fixed-dose rate (FDR) of 10 mg/m^2^/min on days 1 and 8. For the 14-day regimen, G was administered on day 1 by FDR of 10 mg/m^2^/min in approximately 29% of patients, over 90 min in approximately 50%, and over 30 min in approximately 21% of patients, due to the lack of a uniform protocol for all the KPNC medical centers as the 14-day regimen was modified from the 21-day protocol. D was administered over 30 min for both the 14-day (on day 1) and 21-day regimens (on day 8). GCSF: For the 14-day regimen, filgrastim at 5 mcg/kg was given approximately 24 to 48 h after the completion of each chemotherapy for seven days. For the 21-day regimen, filgrastim at 5 mcg/kg was given beginning on day 9 for seven to 10 days. Beginning in March 2020, a majority of the patients was given one dose of pegfilgrastim-jmdb 6 mg by subcutaneous injection instead of filgrastim due to the COVID-19 pandemic.

### 2.3. Assessment for Response, PFS, and OS

The response was assessed according to RECIST version 1.2 criteria [22]. For the 14-day regimen, a majority of the first assessment was performed with either a Computer-aided Topography (CT) or Positron Emission Topography (PET) scan after four cycles (a 14-day regimen was counted as one cycle). For the 21-day regimen, first CT or PET scan was performed after two to three cycles of chemotherapy. The PFS was measured from the date the first cycle of G-D was administered to the date an imaging study showed progression or patient’s physical condition declined significantly and entered comfort care or expired. OS was measured from the date the first cycle of G-D was administered to the date of death.

### 2.4. Next-Generation Sequencing (NGS) for Identifying Genomic Alterations

A majority of the NGS on tumor specimen was performed by StrataNGS (Strata Oncology, Ann Arbor, Michigan). Approximately 10% of the cases were performed using FoundationOne (Foundation Medicine, Cambridge, Massachusetts). StrataNGS interrogated approximately 100 to 400 genes and FoundationOne interrogated approximately 300 to 400 genes [23,24].

### 2.5. Statistical Analysis

The data analysis was performed using MedCalc software (Belgium). The PFS and OS were analyzed using Kaplan–Meyer plot (Log rank test). The statistical comparisons of response rate and toxicities between the two groups of patients were analyzed using the Chi squared method.

## 3. Results

### 3.1. Demographics

We identified 174 patients who were treated with G-D from 1 January 2017 to 30 July 2020 from the KPNC database. Two patients with sarcomatoid carcinoma and 11 patients who received G-D in adjuvant setting were excluded. The total assessable patients were 161. Among these 161 patients, 79 received the 14-day regimen and 82 received the 21-day regimen. The demographics between the two groups were similar and the differences were not statistically significant (Table 1).

The median age for the 14-day regimen was 61, and 60 for the 21-day regimen. The median follow-up was 9.3 months in the 14-day group and 9.5 months in the 21-day group. The most common histology was LMS (39 and 39 patients in each treatment group, including uterine leiomyosarcoma). Other, more common histologies included undifferentiated pleomorphic sarcoma (UPS, eight and 10 patients) and liposarcoma (seven and 10 patients), as well as seven patients with bone sarcoma (osteosarcoma and Ewing’s sarcoma, three and four in each group). There were 21 patients with retroperitoneal sarcoma (11 and 10 in each treatment group). More than 90% of patients received either regimen as a first or second line. Physical performance was ECOG 0 or 1 in 80.5% and 78.5% of patients, respectively, and 2 to 4 in approximately 20% of patients in either group (Table 1). Sixty-seven of 161 (42%) patients had genomic profiling performed, 36 and 31 each in the 14-day and 21-day regimens. The most common genomic alterations were CDKN2A deletion or mutation or CDK4 amplification (*N* = 15), RB1 deletion or mutation (*N* = 18), and TP53 deletion or mutation or MDM2 amplification (*N* = 37). More patients treated with the 14-day regimen had CDKN2A deletion/mutation or CDK amplification than patients treated with 21-day regimen (11 versus 4, *p* = 0.08, Table 1). There were no significant differences on the distribution of RB1 and TP53/MDM2 genomic alterations between the two treatment groups (Table 1).

### 3.2. Gemcitabine and Docetaxel Dosing

For the 14-day regimen, G was given at 1000 mg/m^2^ in 80% of patients, 1250 mg/m^2^ in 2.5% of patients, and 1500 mg/m^2^ in 17.5% of patients, while D was given at 50 mg/m^2^ in 92.5% of patients and 25–40 mg/m^2^ in 7.5% of patients (Table 1). For the 21-day regimen, G was given at 900 mg/m^2^ in 93% of patients and 675 mg/m^2^ or 750 mg/m^2^ in 7% of patients, while D was given at 75 mg/m^2^ in 87.7% of patients, 100 mg/m^2^ in 11% of patients, and 60 mg/m^2^ in 1.2% of patients (Table 1). GCSF was administered to 97% and 100% of patients, respectively (Table 1). For the patients who received G-D as second or later line, the previous chemotherapy was doxorubicin-based 78% and 88%, respectively, in the 14-day and 21-day groups (Table 1).

### 3.3. Lines of Therapy

For the 14-day regimen, 47 patients (59.5%) were treated with G-D as first-line therapy, 27 patients (34.2%) as second-line, and five patients (6.3%) as third line. For the 21-day regimen, 52 patients (63.4%) were treated with G-D as first-line, 22 patients (26.8%) as second-line, and eight patients (9.8%) as third- and fourth-line (Table 1).

### 3.4. Response Rate, PFS and OS

A total of 31 patients obtained a PR or CR, 18 for the 14-day regimen (22.8%) and 13 (15.8%) for the 21-day regimen, which was not statistically significant (*p* = 0.26) (Table 2).

Approximately 27.8% and 29.3% of patients had stable disease (SD), while 49.3% and 52.4% had progressive disease (PD), respectively. Three patients treated with the 14-day regimen obtained CR, including a patient with undifferentiated pleomorphic sarcoma (UPS), a patient with angiosarcoma, and a patient with synovial sarcoma. One patient with relapsed and metastatic intima sarcoma to the adrenal gland treated with the 21-day regimen obtained pathologic CR. The median PFS was 4.0 months (0.03–36.5 months) for the 14-day regimen and 3.2 months (0.3 to 17.7 months) for the 21-day regimen (Table 2 and Figure 1, *p* = 0.15).

The median OS was 12.6 months (0.9 to 39.2 months) for the 14-day regimen and 14.7 months (0.4 to 46.6 months) for the 21-day regimen (Table 2 and Figure 2, *p* = 0.55). There was no significant statistical difference between the two regimens on PFS and OS.

As approximately 60% of patients (59.5% and 63.4% for 14-day and 21-day regimens, respectively) were untreated patients (treated with G-D as first-line therapy), we performed separate analyses to compare if there were significant differences between the two regimens in the first-line setting. The median PFS was 5.2 months for the 14-day regimen, significantly better than 3.2 months for the 21-day regimen (Table 3 and Appendix A, *p* = 0.01). The median OS was not significantly different, with 25.2 months for the 14-day regimen compared to 17.6 months for the 21-day regimen (Table 3 and Appendix A, *p* = 0.57), consistent with that of the entire cohort.

As nearly 50% of the patients treated with either the 14-day or 21-day regimen had advanced or metastatic LMS, we compared the outcomes of LMS patients only. The median PFS was 4.9 and 3.8 months (Table 3 and Appendix A, *p* = 0.01) and the median OS was 29.7 and 16.9 months (Table 3 and Appendix A, *p* = 0.2) for the LMS patients treated with the 14-day and 21-day regimens, respectively, similar with that of the entire patient cohort. For the retroperitoneal sarcoma (total of 21 patients), the median PFS was 4.0 months and 2.1 months and median OS was 14.8 and 12.3 months for 14-day and 21-day regimens, respectively (*p* = 0.80 and 0.71). There were too few patients among the other histology subtypes for meaningfully comparing the outcomes.

### 3.5. Toxicity

Four patients who started with the 21-day regimen switched to the 14-day regimen and eight other patients discontinued G-D chemotherapy completely due to toxicities. Only two patients treated with the 14-day regimen discontinued chemotherapy (Table 2, *p* = 0.006). Ten patients (12.2%) treated with the 21-day regimen developed febrile neutropenia, requiring hospitalization despite being given GCSF, while only one patient treated with the 14-day regimen developed febrile neutropenia (Table 2, *p* = 0.006). There was no significant difference in sensory peripheral neuropathy between the two regimens (Table 2). Anemia (hemoglobin < 10.0/dL) was common with both regimens but a majority was grade 1 or 2. Thrombocytopenia (<100,000/mm^3^) was more common in patients treated with the 21-day regimen and a majority was also grade 1 or 2 (Table 2). Diarrhea was uncommon with either regimen (Table 2). There were no grade 3 gemcitabine-specific toxicities (skin, pulmonary, or other toxicities) that were observed in either regimen.

### 3.6. Association of Genomic Alterations with RR, PFS, and OS

We explored if the common genomic alterations were associated with outcomes (Table 4) [25,26,27,28,29].

TP53 deletion/mutation/MDM2 amplification, CDKN2A deletion/mutation/CDK4 amplification, and RB1 deletion/mutation were the most common genomic alterations. Interestingly, we found that for all the patients with a CDKN2A deletion/mutation or CDK4 amplification, there was absence of RB1 deletion or mutation, and for all the patients whose tumor harbored a RB1 deletion or mutation, there was absence of CDKN2A deletion/mutation or CDK4 amplification, indicating that these two types of genomic alterations are mutually exclusive. Consistent with this finding, 40% of patients with CDKN2A/CDK4 alteration had de-differentiated liposarcoma (26.7% LMS, 20% UPS, and 13.3% osteosarcoma), while 83.3% of patients with a Rb1 alteration had leiomyosarcoma. One out of 15 (6.7%) patients with a CDKN2A deletion/mutation or CDK4 amplification obtained a PR, while 12 out of 52 (23.1%) patients without a CDKN2A deletion/mutation or CDK4 amplification had PR/CR, though not statistically significant (*p* = 0.16). However, patients without a CDKN2A deletion/mutation or CDK4 amplification had nearly statistically significant, longer OS compared to patients with a CDKN2A deletion/mutation or CDK4 amplification (17.6 versus 10.1 months, Figure 3, *p* = 0.06).

Six out of 18 (33.3%) patients with a RB1 deletion/mutation obtained PR/CR, compared to seven out of 49 (14.3%) patients without a RB1 deletion/mutation, though not statistically significant (*p* = 0.08). However, patients with a RB1 deletion/mutation had statistically significant longer PFS compared to patients without a RB1 deletion/mutation (6.2 versus 3.2 months, Figure 4, *p* = 0.05), though the OS was not significantly different. There were no statistically significant differences on RR, PFS, or OS between patients with or without a TP53 deletion/mutation or MDM2 amplification. Other genomic alterations including PTEN, BRCA2, and others were in low frequency not adequate for statistical analysis.

## 4. Discussion

The G-D combination chemotherapy is an effective regimen for patients with locally advanced, relapsed, and/or metastatic STS and bone sarcoma. The SARC002 randomized phase II trial that enrolled the patients who had received zero to three lines of chemotherapy demonstrated superior efficacy with G-D combination (RR 16%, PFS 6 months, and OS 17.9 months) compared to G alone (RR 8%, PFS 3 months, and OS 11.5 months), which further established the G-D combination as a standard first- or second-line chemotherapy option [9]. The GeDDis phase III randomized trial comparing G-D to single-agent doxorubicin in the first-line setting showed that both regimens provided similar efficacy. However, the G-D regimen resulted in higher toxicities. The D in this regimen was 75 mg/m^2^ administered on day 8 and G was 675 mg/m^2^ administered on days 1 and 8, with GCSF support allowed after an episode of febrile neutropenia [14]. The rate of febrile neutropenia and peripheral sensory neuropathy was 12% and 25% in the G-D arm (20% and 11% in the doxorubicin arm). The quality of life assessment 12 weeks post-randomization showed that the C30 scales and FA13 scores favored the doxorubicin arm, allowing the authors to conclude that doxorubicin should remain as the standard, first-line, systemic chemotherapy [14].

More than 90% of our patients were treated with G-D regimen either in the first- or second-line setting and less than 10% of the patients were treated in third- or fourth-line. Our data showed that patients treated with the 14-day regimen, compared to the patients treated with the 21-day regimen, had similar RR, PFS, and OS; a lower rate of febrile neutropenia requiring hospitalization; less-frequent discontinuation of chemotherapy due to intolerance; and fewer patients developed thrombocytopenia. Consistent with the results of the entire cohort, no significant difference on OS was found between the two regimens for the untreated patients (patients treated with G-D regimen as first-line) and for the LMS subset only. This indicates that 14-day G given at 1000 to 1500 mg/m^2^ and D at 50 mg/m^2^ with GCSF support can result in similar outcomes compared to the 21-day regimen given with G at 675–900 mg/m^2^ and D at 75–100 mg/m^2^ with GCSF support, but with an improved toxicity profile. We speculate that the toxicity profile of the 14-day G-D regimen, if compared to single-agent doxorubicin in the first-line setting, is likely to be favorable. This also suggests that if the docetaxel dose in the 21-day regimen is reduced from 75 mg/m^2^ to a lower dose, such as 60 mg/m^2^ (a 20% dose reduction), the toxicity profile could be similar to that of the 14-day regimen, while the efficacy remains similar as well.

G was administered as a 30-minue infusion every 14 days in the Axtell trial [2]. In fact, in a majority of the malignancies (non-small cell lung cancer, breast cancer, transitional cell carcinoma, head and neck cancer, cervical cancer, biliary cancer, etc.) G is administered as a 30-min infusion [30,31]. G was initially thought to be potentially associated with better survival when given by FDR compared to a 30-min infusion in patients with metastatic pancreatic cancer [32]. However, a subsequent Eastern Cooperative Oncology Group (ECOG) trial E6201 showed it was not the case [33]. In fact, some studies have shown that FDR was associated with more hematologic toxicities compared to a 30-min infusion [30,32]. In our study we did not observe G-specific grade 3 toxicities in either the 14-day or 21-day regimen, despite the fact that in the 14-day regimen the administration of G was not uniform across all KPNC medical centers.

Interestingly, our data also showed that the genomic alterations that resulted in the inactivation of CDKN2A were associated with worse OS, consistent with the finding by Bui et al. [19]. In addition, our data showed RB1 deletion or mutation was associated with better PFS. This is consistent with our additional finding that all patients with a RB1 deletion or mutation did not harbor a CDKN2A deletion/mutation or CDK4 amplification and that all patients with a CDKN2A deletion/mutation or CDK4 amplification did not harbor a RB1 deletion/mutation, suggesting that the genomic alterations in these two pathways are mutually exclusive, consistent with the previous study on Rb1 pathway alteration in 31 tumor types using The Cancer Genome Atlas (TCGA) database [34]. However, some of the sarcomas may carry promoter methylation of RB1 and/or CDKN2A, resulting in the inactivation of either gene that is not identified by the NGS [35]. A study showed mutation in INK4A deletion was associated with worse outcomes in Ewing sarcoma [20]. However, it was not reproduced in a Children’s Oncology Group (COG) study [21]. *INK4A* gene locus encodes CDKN2A and p14^AR^F, both of which are tumor suppressor genes [36]. Further studies with a larger patient volume across multiple institutions and specific histology shall provide additional insight.

The limitations of our study include that it was a retrospective and not randomized study, that approximately 20% of patients had poor performance status when initiating G-D and that the documentation of toxicity was likely incomplete. Additionally, due to a small number of non-leiomyosarcoma histology patients, meaningful statistical analysis was not possible to compare the outcomes of the specific histology between the two treatment groups. The strengths of our study are that the number of patients studied in each group was relatively large with similar demographics, and that the majority of patients received D at 50 mg/m^2^ with the 14-day regimen or 75 mg/m^2^ with the 21-day regimen, which is compatible with the widely adopted D dosing in the international sarcoma community.

## 5. Conclusions

In conclusion, our study showed that both 14-day and 21-day G-D regimens provide comparable RR, PFS, and OS, but the 14-day regimen was associated with lower toxicity and better tolerance and can be broadly adopted as a safer and equally effective regimen for patients with metastatic or locally advanced soft tissue and bone sarcoma.

## Figures and Tables

**Figure 1 cancers-13-01983-f001:**
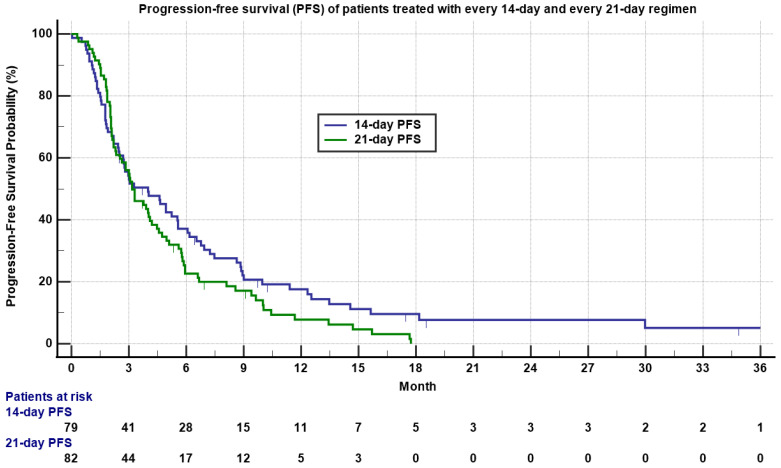
Kaplan–Meyer plot of progression-free survival (PFS) of patients treated with 14-day (**blue**) and 21-day (**green**) regimens. Note median PFS was 4.0 versus 3.2 months for the 14-day regimen and 21-day regimen, respectively (*p* = 0.15). Patients who had not progressed are marked as censored. Number of patients at risk is indicated in the bottom.

**Figure 2 cancers-13-01983-f002:**
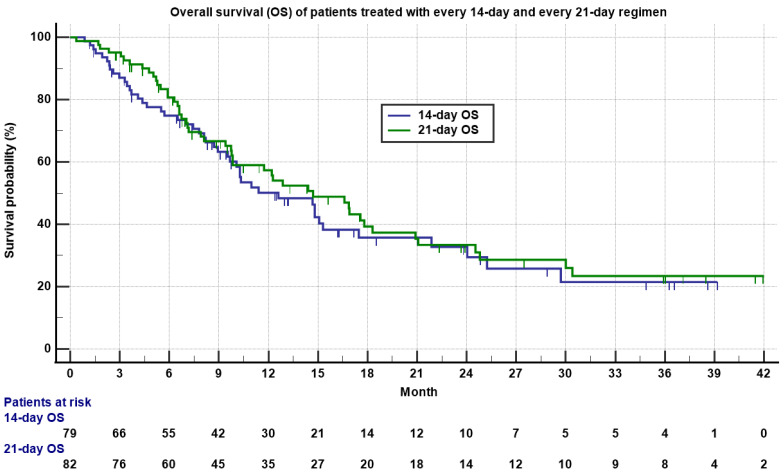
Kaplan–Meyer plot of overall survival (OS) of patients treated with 14-day (**blue**) and 21-day (**green**) regimens. Note the median OS was 12.7 versus 14.6 months for 14-day and 21-day regimens, respectively (*p* = 0.55). Patients who were still alive are marked as censored. Number of patients at risk is indicated in the bottom.

**Figure 3 cancers-13-01983-f003:**
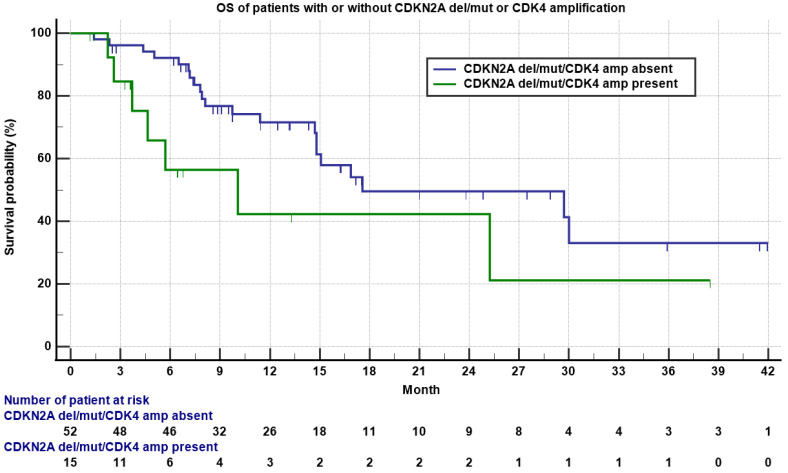
Kaplan–Meyer plot of OS of patients with (**green**) or without (**blue**) a CDKN2A deletion/mutation or CDK4 amplification. Note median OS was 10.1 versus 17.6 months for patients with or without a CDKN2A deletion/mutation or CDK4 amplification (*p* = 0.06). Patients who were still alive are marked as censored. Number of patients at risk is indicated in the bottom.

**Figure 4 cancers-13-01983-f004:**
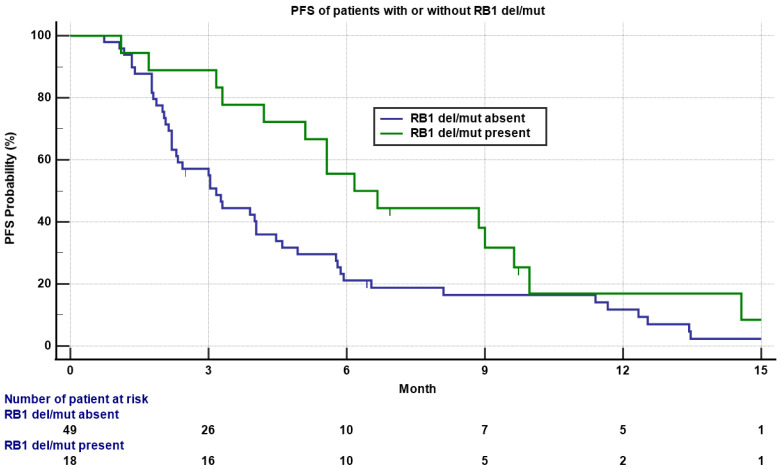
Kaplan–Meyer plot of PFS of patients with (**green**) or without (**blue**) a RB1 deletion/mutation. Note the median PFS was 6.2 versus 3.2 months for patients with or without a RB1 deletion/mutation (*p* = 0.05). Patients who had not progressed are marked as censored. Number of patients at risk is indicated in the bottom.

**Table 1 cancers-13-01983-t001:** Demographics of patients treated with 14-day and 21-day regimens. LMS, leiomyosarcoma; UPS, undifferentiated pleomorphic sarcoma; LPS, liposarcoma; RP sarcoma, retroperitoneal sarcoma.

Demographics			14-Day Regimen (*N* = 79)	21-Day Regimen (*N* = 82)	*p*-Value
Age			61 (25–81)	60 (22–85)	0.92
Sex (Female) (%)			49 (62)	55 (67.1)	0.50
Median follow-up (month)			9.3 (0.1–41)	9.5 (1.5–46)	N/A
Ethnicity	White (%)		40 (50.6)	50 (61)	0.69
	Hispanic (%)		14 (17.7)	21 (25.6)	
	Asian (%)		19 (24.1)	7 (8.5)	
	Black (%)		6 (7.6)	4 (4.9)	
Histology	LMS (%)		39 (49)	39 (47.5)	0.78
	Non-LMS (%)		40 (51)	43 (52.5)	
	Non-LMS subtypes	UPS (%)	8 (10.1)	10 (12.2)	0.82
		LPS (%)	7 (8.9)	10 (12.2)	
		Bone sarcoma (%)	3 (3.4)	4 (4.5)	
		Others (%)	22 (27.8)	19 (23.2)	
	RP sarcoma (%)		11 (13.9)	10 (12.2)	0.75
Median cycles of G-D chemotherapy administered			5 (1–33)	5 (1–18)	N/A
Line of chemotherapy administered	1st (%)		47 (59.5)	52 (63.4)	0.41
	2nd (%)		27 (34.2)	22 (26.8)	
	Later (%)		5 (6.3)	8 (9.8)	
Physical Performance	ECOG 0 to 1		63 (79.7)	64 (78.5)	0.75
	ECOG 2 to 4		16 (20.2)	18 (21.5)	
Gemcitabine	1000 mg/m^2^ (%)		63 (80)		N/A
	1250 mg/m^2^ (%)		2 (2.5)		
	1500 mg/m^2^ (%)		14 (17.5)		
Docetaxel	50 mg/m^2^ (%)		73 (92.5)		
	<50 mg/m^2^ (%)		6 (7.5)		
Gemcitabine	900 mg/m^2^ (%)			75 (91.5)	N/A
	750 mg/m2 (%)			1 (1.2)	
	675 mg/m2 (%)			6 (7.3)	
Docetaxel	60 mg/m^2^ (%)			1 (1.2)	N/A
	75 mg/m^2^ (%)			72 (87.8)	
	100 mg/m^2^ (%)			9 (11.0)	
Filgrastim administered (%)			77 (97.5)	82 (100)	0.15
Prior chemotherapy if 2nd and later line	Doxorubicin-based (%)		62 (78)	72 (88)	0.09
	Others (%)		17 (22)	10 (12)	
Genomic alterations (N = 67)			N = 36	N = 31	
CDKN2A del/mut/CDK4 amp (%)			11 (30.5)	4 (12.9)	0.08
RB1 del/mut (%)			10 (27.8)	8 (25.8)	0.85
TP53 del/mut/MDM2 amp (%)			18 (50)	19 (61.3)	0.36

**Table 2 cancers-13-01983-t002:** Response rate (RR), median progression-free survival (PFS), overall survival (OS), and toxicity profile of patients treated with 14-day and 21-day regimens. A 95% confidence interval (CI) for the median PFS and OS is indicated (inside the parenthesis).

PFS, OS, and Toxicities	14-Day Regimen (*N* = 79)	21-Day Regimen (*N* = 82)	*p*-Value
RR	22.8%	15.8%	0.26
Median PFS (month)	4.0 (0.03–36.5)	3.2 (0.3–17.7)	0.15
Median OS (month)	12.6 (0.9–39.2)	14.7 (0.4–46.6)	0.55
Febrile neutropenia (%)	1 (1.3)	10 (12.2)	0.006
Intolerance (%)	2 (2.5)	12 (14.6)	0.006
Peripheral neuropathy (%)	8 (10.1)	10 (12.2)	0.67
Anemia (hemoglobin < 10/dL)	(%) 45 (57.5)	56 (68.4)	0.15
Thrombocytopenia (<100.000/mm^3^) (%)	7 (8.9)	17 (20.5)	0.03
Diarrhea (%)	5 (6.3)	7 (8.5)	0.74

**Table 3 cancers-13-01983-t003:** Median PFS and OS of patients treated with first-line 14-day and 21-day regimens (all histology), and patients with leiomyosarcoma (LMS) treated with 14-day and 21-day regimens (all lines). A 95% confidence interval (CI) for the median PFS and OS is indicated (inside the parenthesis).

PFS and OS	14-Day Regimen (Month)	21-Day Regimen (Month)	*p*-Value
First-line Median PFS	5.2 (2.7–7.7)	3.2 (2.2–4.0)	0.01
First-line median OS	25.2 (9.5–40.0)	17.6 (7.1–20.9)	0.57
LMS median PFS	4.9 (2.8–8.6)	3.8 (2.1–4.6)	0.01
LMS median OS	29.7 (14.8–29.7)	16.7 (9.8–30.0)	0.2

**Table 4 cancers-13-01983-t004:** RR, median PFS, and median OS by genomic alterations.

*N* = 67		RR	PFS (Month)	OS (Month)
CDK4amp/ CDKN2A del/mut	Yes (*N* = 15)	6.7%	3.0	10.1
	No (*N* = 52)	23.1%	4.9	17.6
*p*-value		0.16	0.37	0.06
RB1 del/mut	Yes (*N* = 18)	33.3%	6.2	17.6
	No (*N* = 49)	14.3%	3.2	16.9
*p*-value		0.08	0.05	0.42
TP53 del/mut, MDM2 amp	Yes (*N* = 37)	13.5%	3.9	16.9
	No (*N* = 30)	26.7%	5.6	25.2
*p*-value		0.18	0.57	0.65

## Data Availability

De-identified data may be available upon approval by KPNC IRB if requested.

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
