# Peer review of "Fourteen-Day Gemcitabine-Docetaxel Chemotherapy Is Effective and Safer Compared to 21-Day Regimen in Patients with Advanced Soft Tissue and Bone Sarcoma"

_cancers, 2021, doi:10.3390/cancers13081983_

Round 1

Reviewer 1 Report

The manuscript reports the retrospective study on the patients with advanced or metastatic soft tissues and bone sarcomas treated with Gemcitabine and Docetaxel combination for the comparison of 14 days regime with 21 days regime. The results show that 14 days regime is as effective as 21 days regime in RR, PFS and OS, while it is less toxic and better tolerated. The study also includes genomic analysis of CDKN2A, CDK4 and RB alterations, and shows that CDKN2A and CDK4 alterations were associated with worse therapeutic outcomes, and RB deletion/mutation was associated with better PFS.

While the study is providing results informative and beneficial to the field of sarcoma clinical management, the authors should correct and improve the manuscript regarding the following points.

-- There is difficulty in studies of clinical trials of sarcoma because of its heterogeneity and rarity. Nevertheless, the authors should consider sarcoma subtypes in the clinical outcome and genomic analysis. While RB genomic alterations in the same histology sarcoma type would be interesting and suggestive of different molecular pathogenesis, the RB deletion/mutation subset may represent a distinct histotype of sarcoma that overall respond better to the chemotherapy. In particular, the tumors with CDK4 amplification with RB-wildtype are likely to include liposarcoma with chromosome 12q13-15 amplification harboring CDK4 and MDM2, which are almost all intact RB and often resistant to chemotherapy. Therefore, the authors should describe histotyps of non-LMS sarcomas used in the study.

-- (p. 2) Interestingly, we have found that genomic alteration of CDKN2A pathway and RB1 deletion/mutation were mutually exclusive which have not been previously reported.

It is well recognized that CDKN2A, CDK4/6 and RB are in the line of same pathway, and thus their genomic alterations are mutually exclusive in many types of cancer including STS. Here’s the recent study reporting TCGA analysis of RB pathway in 31 cancer types. The authors should correct and modify their statement accordingly.

Pan-cancer molecular analysis of the RB tumor suppressor pathway

https://doi.org/10.1038/s42003-020-0873-9

see Figure 1E and Supplementary Fig. 2

Author Response

Reviewer #1 point by point response

Pan et al

The manuscript reports the retrospective study on the patients with advanced or metastatic soft tissues and bone sarcomas treated with Gemcitabine and Docetaxel combination for the comparison of 14 days regime with 21 days regime. The results show that 14 days regime is as effective as 21 days regime in RR, PFS and OS, while it is less toxic and better tolerated. The study also includes genomic analysis of CDKN2A, CDK4 and RB alterations, and shows that CDKN2A and CDK4 alterations were associated with worse therapeutic outcomes, and RB deletion/mutation was associated with better PFS.

While the study is providing results informative and beneficial to the field of sarcoma clinical management, the authors should correct and improve the manuscript regarding the following points.

-- There is difficulty in studies of clinical trials of sarcoma because of its heterogeneity and rarity. Nevertheless, the authors should consider sarcoma subtypes in the clinical outcome and genomic analysis. While RB genomic alterations in the same histology sarcoma type would be interesting and suggestive of different molecular pathogenesis, the RB deletion/mutation subset may represent a distinct histotype of sarcoma that overall respond better to the chemotherapy. In particular, the tumors with CDK4 amplification with RB-wildtype are likely to include liposarcoma with chromosome 12q13-15 amplification harboring CDK4 and MDM2, which are almost all intact RB and often resistant to chemotherapy. Therefore, the authors should describe histotyps of non-LMS sarcomas used in the study.

Response: Thank you for valuing our manuscript. We have attempted to compare the outcomes for other histology subtypes including undifferentiated pleomorphic sarcoma (UPS, N=18) and liposarcoma (N=17), however, due to the number of cases for each subtype is less than 20, it is not statistically feasible to compare (please see Result section 3.4). We have added additional information on the histology subtypes associated with CDKN2A/CDK4 and Rb1 alterations (please see Result section 3.6).

-- (p. 2) Interestingly, we have found that genomic alteration of CDKN2A pathway and RB1 deletion/mutation were mutually exclusive which have not been previously reported.

It is well recognized that CDKN2A, CDK4/6 and RB are in the line of same pathway, and thus their genomic alterations are mutually exclusive in many types of cancer including STS. Here’s the recent study reporting TCGA analysis of RB pathway in 31 cancer types. The authors should correct and modify their statement accordingly.

Pan-cancer molecular analysis of the RB tumor suppressor pathway

https://doi.org/10.1038/s42003-020-0873-9

see Figure 1E and Supplementary Fig. 2

Response: Thank you for pointing this out, we have revised accordingly (please see the last sentence in Introduction section and Reference 34 cited in the Discussion section). This is an excellent point and we appreciate it very much.

Reviewer 2 Report

Comments to the Author

This paper entitled “14-day gemcitabine-docetaxel chemotherapy is effective and safer compared to 21-day regimen in patients with advanced soft tissue and bone sarcoma” is  interesting in evaluating chemotherapy regimens for sarcoma. However, surgery or chemotherapy is the current main treatment for sarcoma, and the evidence is clear. On the other hand, many molecular biological alterations, such as genetic abnormalities and chromosomal translocations, have been discovered in sarcomas, but the effectiveness of therapies targeting them is unknown.This study has been investigated using next-generation sequencing methods, but the relevance of the results is poor and no new findings can be found. We consider it difficult to publish at this state.

Comments

  1. Currently, the multidisciplinary treatment of sarcoma is based on chemotherapy, which is not associated with genetic abnormalities, and although this study examines the effects of chemotherapy and genetic mutations, the basis for this has not been clarified and does not contain any new findings.Deletion of p16 and amplification of MDM2 and CDK4 are also common in the histopathological diagnosis of some sarcomas (liposarcoma).For these reasons, we believe that the conclusions of this study are inadequate for publication.

  1. The authors should re-examine the style of the paper. That is, the abstract should summarize the highlights of the research, not cite other research reports. It should be stated in the introduction.

  1. The analysis of cases with non-constant regimens with different doses per body weight as a group may affect the results.

  1. The importance of dosage, which the authors mention in their discussion, is not emphasized in this study. This is because, for statistical evaluation, drug doses should be compared under the same conditions.

Author Response

Reviewer #2 point by point response

Pan et al.

This paper entitled “14-day gemcitabine-docetaxel chemotherapy is effective and safer compared to 21-day regimen in patients with advanced soft tissue and bone sarcoma” is  interesting in evaluating chemotherapy regimens for sarcoma. However, surgery or chemotherapy is the current main treatment for sarcoma, and the evidence is clear. On the other hand, many molecular biological alterations, such as genetic abnormalities and chromosomal translocations, have been discovered in sarcomas, but the effectiveness of therapies targeting them is unknown.This study has been investigated using next-generation sequencing methods, but the relevance of the results is poor and no new findings can be found. We consider it difficult to publish at this state.

Comments

  1. Currently, the multidisciplinary treatment of sarcoma is based on chemotherapy, which is not associated with genetic abnormalities, and although this study examines the effects of chemotherapy and genetic mutations, the basis for this has not been clarified and does not contain any new findings.Deletion of p16 and amplification of MDM2 and CDK4 are also common in the histopathological diagnosis of some sarcomas (liposarcoma).For these reasons, we believe that the conclusions of this study are inadequate for publication.

 Response: Our main point in this manuscript is to describe the similar outcomes between two different regimens: 14-day and 21-day gemcitabine-docetaxel regimens and that the 14-day regimen is less toxic and safer. We believe these results are new and will be helpful to the oncology community for providing treatment for patients with sarcoma. We explored the genomic alterations and the associated outcomes which adds additional information to the literature.

  1. The authors should re-examine the style of the paper. That is, the abstract should summarize the highlights of the research, not cite other research reports. It should be stated in the introduction.

Response: We have revised as suggested. Thank you.

  1. The analysis of cases with non-constant regimens with different doses per body weight as a group may affect the results.

Response: The main point of our manuscript is to determine the two different regimens with different dosing and schedule resulted in similar outcomes yet different severity of toxicities. Please see Table 2.

  1. The importance of dosage, which the authors mention in their discussion, is not emphasized in this study. This is because, for statistical evaluation, drug doses should be compared under the same conditions.

Response: In the Result section 3.2, we described in details of the dosing differences for the two regimens. More than 90% of patients in both regimens received standard doses which makes our comparison valid. It would be not possible to compare outcomes with every different dosing level because the number of cases would be too few to compare in those patients who received dose reduction to begin with.

Reviewer 3 Report

The authors described the retrospective clinical data of gemcitabine-docetaxel combination to soft tissue sarcoma patients, comparing 14-day regimen to 21-regimen. The appropriate dose setting of gemcitabine-docetaxel combination is still in controversy yet, so the manuscript would help information us considering the clinical problem.

I would like state some points to be improved.

  1. In this analysis, there were similar number of patients with 14-day regimen and 21-day regimen, but the process of choosing regimens to each patients was not unclear. If the regimen was chosen by patients' preference or physicians' choice, the process should be addressed in the "2. Patients and Methods" section. The lack of randomization should be also described as the limitation of the study in the "4. Discussion" section.
  2. The toxicity data of each regimen was only described as febrile neutropenia, peripheral neuropathy in the table 2. I know that these are important events in comparing two schedules of the regimens, but other major adverse events such as anemia, thrombocytopenia or diarrhea should be shown in the table.

Author Response

Point by point response to Reviewer #3

Pan et al

The authors described the retrospective clinical data of gemcitabine-docetaxel combination to soft tissue sarcoma patients, comparing 14-day regimen to 21-regimen. The appropriate dose setting of gemcitabine-docetaxel combination is still in controversy yet, so the manuscript would help information us considering the clinical problem.

Response: Thank you for valuing our manuscript, we are very much appreciative.

I would like state some points to be improved.

  1. In this analysis, there were similar number of patients with 14-day regimen and 21-day regimen, but the process of choosing regimens to each patients was not unclear. If the regimen was chosen by patients' preference or physicians' choice, the process should be addressed in the "2. Patients and Methods" section. The lack of randomization should be also described as the limitation of the study in the "4. Discussion" section.

Response: We have added to the manuscript this information as suggested. The choice of regimen is determined by treating oncologist (please see section 2.1 and the Discussion section). Thank you.

  1. The toxicity data of each regimen was only described as febrile neutropenia, peripheral neuropathy in the table 2. I know that these are important events in comparing two schedules of the regimens, but other major adverse events such as anemia, thrombocytopenia or diarrhea should be shown in the table.

Response: We have added the data as suggested, please see Table 2 and Result section 3.5. Thank you.

Round 2

Reviewer 1 Report

The authors reasonably made changes and improved the revised manuscript.

Author Response

please see email communications

Reviewer 2 Report

This revisit has not been fully vetted in terms of experimental design and evaluation methods. It is difficult to understand the purpose and benefits of this study, and there seems to be a lack of new findings. The lack of changes compared to the first version of the manuscript makes it difficult to evaluate further.

Author Response

please see email communicaitons
